# Improvement of Fertilization Capacity and Developmental Ability of Vitrified Bovine Oocytes by JUNO mRNA Microinjection and Cholesterol-Loaded Methyl-β-Cyclodextrin Treatment

**DOI:** 10.3390/ijms24010590

**Published:** 2022-12-29

**Authors:** Xi Xu, Tong Hao, Emma Komba, Baigao Yang, Haisheng Hao, Weihua Du, Huabin Zhu, Hang Zhang, Xueming Zhao

**Affiliations:** Institute of Animal Sciences (IAS), Chinese Academy of Agricultural Sciences (CAAS), Beijing 100193, China

**Keywords:** JUNO, oocytes, vitrification, bovine, microinjection

## Abstract

Vitrification of oocytes is crucial for embryo biotechnologies, germplasm cryopreservation of endangered and excellent female animals, and the fertility of humans. However, vitrification significantly impairs the fertilization ability of oocytes, which significantly limits its widely used application. JUNO protein, a receptor for Izumo1, is involved in sperm-oocyte fusion and is an indispensable protein for mammalian fertilization, and its abundance is susceptible to vitrification. However, it is still unclear how vitrification reduces the fertilization capacity of bovine oocytes by affecting JUNO protein. This study was designed to investigate the effect of vitrification on the abundance and post-translational modifications of JUNO protein in bovine oocytes. Our results showed that vitrification did not alter the amino acid sequence of JUNO protein in bovine oocytes. Furthermore, the liquid chromatography-tandem mass spectrometry (LC-MS/MS) analysis results showed that vitrification significantly reduced the number and changed the location of disulfide bonds, and increased the number of both phosphorylation and glycosylation sites of JUNO protein in bovine oocytes. Finally, the fertilization capacity and development ability of vitrified oocytes treated with 200 pg JUNO mRNA microinjection and cholesterol-loaded methyl-β-cyclodextrin (CLC/MβCD) were similar to those of fresh oocytes. In conclusion, our results showed that vitrification of bovine oocytes did not alter the protein sequence of JUNO, but induced post-translational modifications and changed protein abundance. Moreover, the fertilization and development ability of vitrified bovine oocytes were improved by the combination treatment of JUNO mRNA microinjection and CLC/MβCD.

## 1. Introduction

Oocyte cryopreservation provides sufficient material for animal biotechnologies to break the limitation of time and space [1,2,3], which can facilitate the worldwide spread of precious female genetics for commercial purposes [4,5]. Moreover, oocyte cryopreservation provides the opportunity to keep the safety of oocytes from epidemic diseases [6] and it is more convenient and economical than protecting the living organism [7], which can be used for gene banking of female animals to store genetic resources [8]. For humans, oocyte cryopreservation allows women to delay reproduction, during professional development, and enables cancer patients to maintain fertility [9,10].

As previously reported, oocytes are primarily sensitive to cooling due to their high water content, large size, and unique cell structure [11]. The process of cryopreservation may cause a series of cryoinjuries, which decrease the maturation rate of oocytes [12] and hinder their developmental potential [13], especially their ability to fertilize [14]. Compared with fresh oocytes, the fertilization rate of frozen-thawed oocytes decreased significantly [14,15,16,17,18]. The fertilization rate of mouse oocytes frozen by standard slow freezing (65.5 ± 2.1%) was significantly lower than that of fresh oocytes (90.3 ± 1.6%) [14]. The cleavage and blastocyst rates of vitrified oocytes after fertilization were significantly lower than those of the fresh group in sheep and bovine [19,20], which seriously limited the utilization of vitrified oocytes.

It has been proven that CD9 [21] and JUNO [22], located in oocytes, and Izumo1 [23], located in sperm, are crucial to fertilization. As a member of the tetraspanin superfamily of proteins, CD9 plays a vital role in sperm-oocyte interactions during fertilization [21]. Loss of CD9 function reduces oocyte fertilization rates in mice [24], pigs [25], and bovines [26]. As previously reported, Izumo1 is the first sperm membrane protein indispensable for mouse sperm-oocyte fusion [23]. Izumo1 knockout male mice are infertile because of the failure of sperm-oocyte fusion [23]. However, Inoue [27] reported that the binding of Izumo1 to the surface of oocytes may not need CD9, and oocytes lacking CD9 can be fertilized, which suggests that CD9 is not the only receptor for Izumo1.

Like the Izumo1 receptor, JUNO is a glycosylphosphatidylinositol (GPI)-anchored protein expressed on the oocyte surface [22]. JUNO-Izumo1 interaction is an important adhesion event between sperm and oocyte, which is necessary for fertilization [22]. Moreover, JUNO is rapidly shed from the oocyte membrane within vesicles after normal fertilization, providing a potential mechanism for the membrane block to avoid polyspermy in mammals [22]. As previously reported, JUNO protein abundance of oocytes is positively correlated with the capacity for fertilization in oocytes [28,29,30], and the oocytes lacking JUNO fail to fuse with normal sperm [31]. Izumo1-JUNO interactions are conserved within mammals, such as sheep [32], cattle [33], opossums, pigs, and humans [22]. As previously reported, vitrification significantly reduces the abundance of JUNO and the fertilization ability of bovine oocytes [34]. However, the mechanism by which vitrification affects the JUNO protein function in bovine oocytes remains unclear, and efficient methods are urgently required to improve the fertilization ability of vitrified bovine oocytes.

Therefore, the present study was designed to investigate the effects of vitrification on the expression abundance and post-translational modifications of JUNO protein in bovine oocytes. Then, the sperm binding number, fertilization ability, and development ability of vitrified bovine oocytes after JUNO mRNA microinjection were examined. Finally, the effects of the combination treatment of JUNO mRNA microinjection and CLC/MβCD on the fertilization and developmental capacity of vitrified bovine oocytes were examined, as well as the quality of their blastocysts.

## 2. Results

### 2.1. Effect of Vitrification on JUNO Protein Abundance in Bovine Oocytes

Figure 1A shows the representative images of JUNO staining. As shown in Figure 1B, the JUNO protein abundance of the vitrification group was significantly lower than that of the fresh group (*p* < 0.05).

### 2.2. Effect of Vitrification on the Post-Translational Modifications of JUNO Protein Sequence in Bovine Oocytes

As shown in Figure 2, 95.5% (232) of all amino acids (243) of JUNO proteins were detected in both fresh and vitrification groups. No difference was found in the 232 amino acids between the fresh and vitrification groups.

Furthermore, the disulfide bonds, phosphorylation, and glycosylation modification of JUNO were different between the fresh group and the vitrification group. All the mass spectrograms are shown in Appendix A. As shown in Table 1, eight disulfide bonds were detected in the fresh group, located at 18–179, 18–165, 27–125, 79–47, 79–79, 79–99, 95–165, and 99–165; and five disulfide bonds were detected in the vitrification group, located at 79–79, 79–179, 79–229, 95–95, and 125–142. Only one disulfide bond was detected in both the fresh group and the vitrified group (79–79). As shown in Figure 2, only one phosphorylation site (75) was detected in the vitrified group, while no phosphorylation modification was detected in the fresh group. Moreover, there glycosylation sites were detected in the vitrification group, located at 174, 184, and 225; only one glycosylation site was detected in the fresh group, located at 174.

### 2.3. Effect of JUNO mRNA Microinjection on the Protein Abundance of JUNO in Vitrified Oocytes

The prepared JUNO mRNA was transfected into 293T cells and the abundance of JUNO protein in 293T cells was verified by western blot analysis. As shown in Figure 3, the expression of JUNO protein in the transfected JUNO mRNA group was similar to that of the pcDNA3-JUNO-Flag group, and the molecular weight of JUNO was 30 kDa. In contrast, the molecular weight of the reference gene(action) was 42 kDa.

Figure 4A shows that the protein abundance of the 200 pg JUNO mRNA fresh group was significantly higher than that of the fresh group. Correspondingly, the JUNO protein abundance of each JUNO mRNA vitrification group was significantly higher than that of the vitrification group but lower than that of the fresh group (*p* < 0.05). On the other hand, the JUNO protein abundance of the 100 pg JUNO mRNA vitrification group was similar to that of the 300 pg JUNO mRNA vitrification group, but significantly lower than that of the 200 pg JUNO mRNA vitrification group (*p* < 0.05).

As shown in Figure 4B, the sperm binding capacity of the 100 pg (12.32 ± 1.16), 200 pg (18.78 ± 1.37), and 300 pg (13.19 ± 1.16) JUNO mRNA vitrification groups were significantly higher than that of the vitrification group (9.53 ± 0.97, *p* < 0.05), and lower than that of the fresh group (22.76 ± 2.18, *p* < 0.05) and the 200 pg JUNO mRNA fresh group (31.09 ± 2.83, *p* < 0.05). Moreover, the sperm binding capacity of the 200 pg JUNO mRNA vitrification group was significantly higher than that of the 100 pg (*p* < 0.05) and 300 pg (*p* < 0.05) JUNO mRNA vitrification groups.

As shown in Figure 4C, the normal fertilization rate of the 100 pg (67.24 ± 5.81%), 200 pg (76.19 ± 6.75%), and 300 pg (66.67 ± 5.19%) JUNO mRNA vitrification groups were significantly higher than that of the vitrification group (57.62 ± 4.87%, *p* < 0.05), but significantly lower than that of the fresh group (84.21 ± 7.39%, *p* < 0.05) and the 200 pg JUNO mRNA fresh group (94.23 ± 8.31%, *p* < 0.05). The normal fertilization rate of the 200 pg JUNO mRNA vitrification group was significantly higher than that of the 100 pg (*p* < 0.05) or 300 pg (*p* < 0.05) JUNO mRNA vitrification groups.

### 2.4. Effect of JUNO mRNA Microinjection on the Development Ability of Vitrified Oocytes

As shown in Table 2, the cleavage and blastocyst rates of the 200 pg JUNO mRNA vitrification group were significantly higher than those of the vitrification group (*p* < 0.05) and lower than those of the fresh group (*p* < 0.05) and the 200 pg JUNO mRNA fresh group (*p* < 0.05).

### 2.5. Effect of the JUNO mRNA + CLC/MβCD Treatment on the Membrane Integrity of Vitrified Oocytes

Figure 5A shows the representative pictures of Annexin V staining. As shown in Figure 5B, the rate of oocytes with PS externalization events in the vitrification group of 200 pg JUNO mRNA + CLC/MβCD treatment (10.94 ± 0.67%) was significantly lower than that in the vitrified group (38.60 ± 2.41%, *p* < 0.05), and similar to that of the fresh group (10.42 ± 0.98%, *p* > 0.05).

### 2.6. Effect of the JUNO mRNA + CLC/MβCD Treatment on the Gene Expression of Vitrified Oocytes

As shown in Figure 6, mRNA expression levels of *ATP6*, *BCL2*, and *SOD1* genes of the 200 pg JUNO mRNA + CLC/MβCD vitrification group were similar to those of the fresh group (*p* > 0.05), and significantly higher than those of the 200 pg JUNO mRNA vitrification group and the vitrification group (*p* < 0.05). The mRNA expression level of *BAX* gene of the 200 pg JUNO mRNA + CLC/MβCD vitrification group was similar to that of the fresh group (*p* > 0.05) and significantly lower than those of the 200 pg JUNO mRNA vitrification group and the vitrification group (*p* < 0.05).

### 2.7. Effect of the JUNO mRNA + CLC/MβCD Treatment on the JUNO Protein Abundance and Sperm Binding Number of Vitrified Oocytes

As shown in Figure 7A, the JUNO protein abundance of the fresh group was significantly higher than that of the vitrification group, but lower than that of the 200 pg JUNO mRNA + CLC/MβCD vitrification group (*p* < 0.05). As shown in Figure 7B, the same results were found in the sperm binding capacity (21.41 ± 2.04, 8.95 ± 0.78, 31.08 ± 2.92, and *p* < 0.05).

### 2.8. Effect of JUNO mRNA + CLC/MβCD Treatment on the Development Ability of Vitrified Oocytes after IVF

As shown in Table 3, the cleavage rate, blastocyst rate, and cell number per blastocyst of the 200 pg JUNO mRNA + CLC/MβCD vitrification group were significantly higher than those of the vitrification group (*p* < 0.05) and similar to those of the fresh group (*p* > 0.05).

### 2.9. Effect of JUNO mRNA + CLC/MβCD Treatment on the Gene Expression in Vitrified Blastocysts

As shown in Figure 8, mRNA expression levels of *IFN-tau*, *BCL2*, *OCT4*, and *CDX2* genes of the fresh group were significantly higher than those of the vitrification group (*p* < 0.05), but significantly lower than those of the 200 pg JUNO mRNA + CLC/MβCD vitrification group (*p* < 0.05). However, the mRNA expression level of the *BAX* in the fresh group was similar to that of the JUNO mRNA + CLC/MβCD vitrification group (*p* > 0.05), but significantly lower than that of the vitrification group (*p* < 0.05).

## 3. Discussion

Previous studies in mice have found that BAP [29], melamine [35], hydroxyurea [30], bis(2-Ethylhexyl) phthalate [36], and senility [28] can significantly decrease the abundance of JUNO protein in oocytes. As shown in Figure 1B, vitrification significantly decreased the abundance of JUNO protein in bovine oocytes, which may be due to the altered expression of *TET1* and *DNMT1* in oocytes induced by vitrification [34,37,38]. TET1 [39] and DNMT1 [40] are essential enzymes that regulate gene methylation levels. DNA methylation can regulate intragenic promoter activity [41], which is essential for oocytes [42] and embryos [40]. The JUNO protein, a receptor for IZUMOL protein, has been proven to be essential for successful fertilization [22]. Therefore, our results suggest that vitrification affects oocyte fertilization capacity by affecting JUNO protein abundance.

Besides, in order to investigate the effect of vitrification on the protein post-translational modifications, our experiment utilized LC-MS/MS to investigate JUNO protein. As shown in Figure 2, for the 243 amino acids of JUNO protein, 95.5% (232) amino acids were detected in both the fresh and the vitrification groups. Furthermore, the sequence of these 232 amino acids was the same in these two groups, indicating that vitrification did not disturb the amino acid sequence of JUNO protein. It is well known that post-translational modifications of proteins are essential for maintaining the normal physiological functions of proteins. These events affect protein stability, localization, function, and participation in different biological processes [43,44,45]. In females, a large number of proteins possessing post-translational modifications, such as hCG [46], FSH [47], BMP15 [48], and GDF9 [49], are crucial for female fertility. JUNO and IZUMO1 are essential for fertilization and adhesion of gamete cell membranes [22], and IZUMO1 interacts with JUNO via its N-terminal domain [50]. Therefore, the differences in the disulfide bond, phosphorylation, and glycosylation of JUNO proteins between vitrified and fresh oocytes were compared in our experiment.

As shown in Table 1, five and eight disulfide bonds were found in the vitrified and the fresh group, respectively. Locations of disulfide bonds in the vitrified group differed from those in the fresh group, except for the disulfide bond located at 79–79. As previously reported, the disulfide bond plays a vital role in fertilization [51,52,53,54]. Our results suggest that vitrification may affect fertilization by altering the number and location of disulfide bonds of JUNO proteins. Similar to our study, previous studies reported that disulfide bonds play an important role in physiological processes associated with fertilization, such as zonal pellucida hardening [55,56].

As shown in Figure 2, vitrification increased the phosphorylation modification of the JUNO protein in bovine oocytes. It is known that phosphorylation acts as a button to turn protein activity on or off [57], and the phosphorylation level of proteins is essential for fertilization [58]. Previous studies have shown that the phosphorylation of H2AX protein can be affected by freeze-thawing [59], and the regulation of phosphorylation of DRP1 can enhance the in vitro development of vitrified mouse GV oocytes [60]. In embryos, the phosphorylation level of protamine catalyzed by SRPK1 is important for early embryonic development [61]. Besides, phosphorylation of tyrosine residues of proteins is also important for activities related to fertilization [62], such as ZP binding and the binding and fusion of sperm-oocyte [63].

Meanwhile, the number of glycosylation sites of JUNO was higher in the vitrified group than that in the fresh group, as shown in Figure 2. In females, protein N-glycosylation in oocytes is important for fertility, as it controls the development of the oocytes [64]. For example, DPAGT1 is involved in the first process of protein N-glycosylation, and its missense mutation results in the poor development of mice oocytes [64]. Meanwhile, glycosylation is essential for normal fertilization [65], and O-glycans partially mediate the properties of ZP3 [66]. Furthermore, N-linked chains of zona glycoproteins play an essential role in species-selective recognition in cattle and pigs [67].

Our study revealed that vitrification affected the abundance and post-translational modifications of JUNO. To the best of our knowledge, this was the first report about the effect of vitrification on JUNO protein modifications of mammalian oocytes, which contributed to illustrating the fundamental effect of vitrification on oocytes.

It has been proven that JUNO protein abundance could be regulated in mammalian oocytes [30,68], and that JUNO protein abundance in oocytes is positively correlated with their fertilization ability during IVF [28,29,36,69]. Therefore, the mRNA of the *JUNO*(*FOLR4*) gene was synthesized in vitro and microinjected into the GV oocytes. As shown in Figure 3, the prepared JUNO mRNA could be typically expressed in 293T cells, and its molecular weight was 30 kDa, which suggests that the JUNO mRNA could be used to regulate the JUNO protein abundance in bovine oocytes.

As shown in Figure 4A, 200 pg JUNO mRNA microinjection successfully increased the JUNO abundance of bovine oocytes. This result confirmed that the protein abundance could be improved by microinjection of the appropriate amount of mRNA [70]. Meanwhile, 100 pg JUNO mRNA microinjection was not enough to protect the JUNO abundance in vitrified bovine oocytes, and 300 pg JUNO mRNA microinjection may lead to too much damage of the oocytes [71]. As shown in Figure 4B, our results indicated that the sperm binding capacity of the 200 pg JUNO mRNA fresh group was significantly higher than that of the fresh group, indicating that JUNO mRNA microinjection at the GV stage successfully increases the JUNO abundance and function. Similar to the previous study [30], the enhancement of JUNO abundance and function significantly increased the normal fertilization rate (Figure 4C), which was due to the ability of JUNO to bind to Izumo1, to promote the adhesion and fusion of sperm-oocyte [72]. Our results also showed that the JUNO protein abundance, sperm binding capacity and normal fertilization rate of the 200 pg JUNO mRNA vitrification group were higher than those of the vitrification group, but still lower than those of the fresh group, indicating that the up-regulation of JUNO protein levels before vitrification was not enough to maintain the normal JUNO protein abundance and function.

Subsequently, the effect of JUNO mRNA microinjection on the development ability of vitrified bovine oocytes was examined. As shown in Table 2, 200 pg JUNO mRNA microinjection at the GV stage could increase the development ability of bovine oocytes. Similar to our results, Sui [73] reported that mouse oocytes with higher JUNO abundance possessed higher cleavage and blastocyst rates, which was due to the increased JUNO abundance which can promote the fertilization [28]. The development ability of the 200 pg JUNO mRNA vitrification group was lower than that of the fresh group, indicating that up-regulation of JUNO protein abundance in oocytes before vitrification may not be enough to restore the development ability in vitrified bovine oocytes.

The damage to the oocyte membrane is one of the significant damages caused by vitrification [74]. Cholesterol is a major structural lipid constituent of the membrane, and plays a vital role in resisting low temperatures [75]. As a cyclic oligosaccharide, MβCD can load or remove cholesterol to the cell membrane [76], which has been proven in bovine oocytes [77]. It has been proven effective in reducing membrane damage and improving the development ability of vitrified oocytes, by increasing the cholesterol level before vitrification and removing the added cholesterol after warming, to restore the original level [78]. Therefore, our study tested whether vitrified bovine oocyte quality and developmental capacity could be improved through treatment with JUNO mRNA + CLC/MβCD.

PS flips from the inner membrane to the outer membrane when apoptosis or death occurs, so PS externalization is a popular marker for apoptosis [79]. As shown in Figure 5B, the combination treatment of JUNO mRNA + CLC/MβCD effectively reduced apoptosis of vitrified bovine oocytes. This result was due to the ability of cholesterol to reduce apoptosis-like changes [80].

The gene expression levels in oocytes were detected to investigate the effect of JUNO mRNA + CLC/MβCD on oocyte quality (Figure 6). The expression levels of *ATP6* and *ATP8* genes are used as indicators of mammalian oocytes [81] and embryo quality [82], and *SOD1* is one of the genes involved in antioxidant activity in oocytes [83,84]. It is well known that *BAX* is a pro-apoptotic gene, while *BCL2* is an anti-apoptotic gene [85]. Therefore, our results suggest that the treatment of JUNO mRNA + CLC/MβCD can improve the quality of vitrified oocytes by upregulating the expression of *ATP6*, *BCL2*, and *SOD1* genes, and downregulating the expression of *BAX* gene.

As shown in Figure 7B, the treatment of JUNO mRNA + CLC/MβCD significantly improved the fertilization ability of vitrified oocytes. Firstly, cholesterol supplementation reduced the apoptotic damage caused by vitrification and protected the integrity of the membrane [86]. Secondly, JUNO mRNA microinjection significantly improved the vitrified oocytes’ JUNO abundance and function. JUNO is essential for fertilization, and it is a GPI-anchored protein enriched in non-invaginated membrane rafts [22]. The membrane raft integrity is essential for accomplishing fertilization in the mouse oocyte, and cholesterol could improve the fertilization ability of the mouse oocyte by protecting the rafts [87]. It has been proven that the localization and level of rafts at the plasma membrane were distinctly affected by vitrification, which can be recovered by adding cholesterol before vitrification and removing it after thawing [78]. All of the evidence contributed to explaining the increased abundance of JUNO protein and sperm binding number in the JUNO mRNA + CLC/MβCD group.

Moreover, as shown in Table 3, the combined treatment of JUNO mRNA + CLC/MβCD improved the development ability of vitrified oocytes, which was attributed to the following two reasons. The first reason was the increased JUNO protein abundance in vitrified oocytes from the JUNO mRNA + CLC/MβCD group, which increased the cleavage rate by increasing the normal fertilization rate. The second reason was that CLC could improve the quality of oocytes and reduce the membrane damage and apoptosis caused by vitrification [78,88]. Similar to our results, Sprícigo [89] reported an increased proportion of oocytes reaching the MII stage and the blastocyst rate at Day7 following exposure to MβCD before vitrification.

*IFN-tau* is the key molecule involved in maternal recognition of pregnancy in bovines [90]. *BAX* is a pro-apoptotic gene, while *BCL2* is an anti-apoptotic gene [85]. *OCT4* is the key factor in maintaining the pluripotency of *ICM* in bovine blastocysts [91]. *SOX2* is closely related to the total cell number of blastocysts [92]. Therefore, the expression levels of *IFN-tau*, *BAX*, *BCL2*, *OCT4*, and *SOX2* have been utilized to assess the quality of blastocysts [93]. As shown in Figure 8, our results showed that mRNA expression levels of *IFN-tau*, *BCL2*, *OCT4*, and *CDX2* in blastocysts from the JUNO mRNA + CLC/MβCD group were significantly increased, and the mRNA expression level of BAX was significantly reduced, indicating that the quality of blastocysts from the JUNO mRNA + CLC/MβCD vitrification group was significantly improved. The higher quality of blastocysts from the JUNO mRNA + CLC/MβCD vitrification group could be explained by CLC because it can reduce membrane damage and apoptosis during vitrification [78].

## 4. Materials and Methods

Unless expressly stated, all chemicals used in this study were purchased from Sigma-Aldrich Chemical Company (St. Louis, MO, USA), and plastic items were purchased from Thermo Fisher Scientific Company (Waltham, MA, USA).

### 4.1. In Vitro Maturation of Oocytes (IVM)

Bovine ovaries from a local slaughterhouse were placed in a physiological saline solution containing streptomycin and penicillin at 35 °C and delivered to the laboratory within 2 h. Cumulus-oocyte complexes (COCs) were collected from 2–8 mm diameter follicles. COCs with more than three layers of intact cumulus cells were used for IVM. The IVM medium consisted of M199 (Gibco BRL, Grand Island, NY, USA) supplemented with 10 μg/mL follicle-stimulating hormone (FSH), 10 μg/mL luteinizing hormone, 1 μg/mL estradiol, and 10% (*v*/*v*) fetal bovine serum (FBS; Gibco BRL, Grand Island, NY, USA). For IVM, 50 COCs were cultured in 500 μL of IVM medium, in 4-well dishes containing mineral oil at 38.5 °C, with 5% CO_2_ for 22–24 h.

### 4.2. Oocytes Vitrification and Warming

MII oocytes were vitrified using the OPS method [94]. Briefly, oocytes were incubated in 10% (*v*/*v*) dimethyl sulfoxide (DMSO) and 10% (*v*/*v*) ethylene glycol (EG) in DPBS containing 3 mg/mL BSA for 30 s, followed by exposure to 15% (*v*/*v*) DMSO and 15% (*v*/*v*) EG in DPBS containing 0.5 M sucrose, 300 g/L Ficoll, and 20% (*v*/*v*) FBS. Then, they were loaded into an OPS and immediately plunged into liquid nitrogen (LN_2_) in a vertical orientation in 25 s.

The OPS was taken from LN_2_ for thawing, and the thin end containing oocytes was dipped into a 0.5 M sucrose solution. Then, oocytes were expelled from the straw and incubated in 0.5 M sucrose for 5 min. The thawed oocytes with a homogenous cytoplasm and good membrane integrity were selected for the subsequent experiments.

### 4.3. In Vitro Fertilization Experiments (IVF)

The IVF procedure was performed according to the method described by Brackett [95], with minor changes. Briefly, one straw of frozen semen (Beijing Dairy Cattle Center, Beijing, China) was removed from LN_2_, plunged into a 38 °C water bath, and washed twice in the 7 mL Brackett and Oliphant medium (BO medium) by centrifugation at 1500 rpm for 5 min. The sperm was added to the fertilization medium (BO medium containing 20 μg/mL heparin, 20 mg/mL BSA, 100 IU/mL penicillin and 100 μg/mL streptomycin), and the final concentration of sperm was adjusted to 1 × 10^6^/mL. Then, 20–30 oocytes were placed in 100 μL fertilization drops for insemination at 38.5 °C with 5% CO_2_. After 16–18 h of fertilization, presumptive zygotes were transported to a CR1aa medium containing 6 mg/mL BSA for 48 h, and then cultured in a CR1aa medium supplemented with 10% FBS for 5 days, with half of the medium replaced every 48 h.

### 4.4. Assessment of JUNO Protein Abundance in Bovine Oocytes

According to a previous method [22], the abundance of JUNO protein in oocytes was examined. Briefly, oocytes were placed in 4% paraformaldehyde for 30 min, transferred to 0.5% Triton X-100 for 20 min, then treated in 1% BSA for 1 h. Subsequently, oocytes were incubated with JUNO antibody (125107, 1:1000 dilution; BioLegend, San Diego, CA, USA) at 4 °C overnight. The negative control group was incubated with 1% BSA instead of an antibody. Finally, the epifluorescence inverted microscope (TE2000-U; Nikon, Tokyo, Japan), connected to a DSRi1 camera (Nikon, Tokyo, Japan), was used to take images of oocytes along the equatorial plane of the oocyte at 10× magnification. Fluorescent images were analyzed by Image J software [69].

### 4.5. JUNO Protein Sequencing and Protein Modification Analysis Based on LC-MS/MS

Oocytes from the vitrification and fresh groups were collected and stored in LN_2_ before further use. All the protein sample preparation and data analysis for LC-MS/MS were performed by Biotech Pack Scientific Cooperation (Beijing, China). JUNO protein sequencing and protein modification analysis, based on LC-MS/MS, was performed according to the method described by the manufacturer’s instructions. In brief, JUNO proteins were obtained from oocytes using RIPA Lysis and Extraction Buffer (ThermoFisher, Waltham, MA, USA), Pierce™ BCA Protein Assay Kit (ThermoFisher, Waltham, MA, USA), and Pierce Co-Immunoprecipitation (Co-IP) Kit (ThermoFisher, Waltham, MA, USA), according to the manufacturer’s instruction. Subsequently, JUNO proteins were separated by SDS-PAGE electrophoresis with a concentration of 12% and electrophoresis parameters of 80 kV for 15 min, followed by 120 kV for 1 h. The obtained JUNO protein was treated with Trypsin (Promega, Madison, WI, USA), Chymotrypsin (Promega, Madison, WI, USA), Pepsin (Promega, Madison, WI, USA), Trypsin & Glu-C (Promega, Madison, WI, USA), and Trypsin & ASP-N (Promega, Madison, WI, USA). The disulfide bonds of JUNO were analyzed using pLink. The phosphorylation and glycosylation were analyzed by Byonic. JUNO protein phosphorylation and glycosylation sites were selected with a Byonic score ≥ 100 because a minimum Byonic score of 100 is considered as highly confident assignment [96].

### 4.6. Preparation and Validation of JUNO mRNA Used for Microinjection

The sequence of JUNO mRNA was downloaded from NCBI (Accession No. XM_024975624.1). Then, pUC57 (Genewiz, Tianjin, China) and c-flag-pcDNA3 (Addgene#20011) vectors were used to construct the JUNO mRNA transcription template. The cDNA was synthesized on the pUC57 vector, then cut from the pUC57 vector and connected to the c-flag-pCDNA3 vector to construct JUNO mRNA transcription template. The DNA template of JUNO mRNA was amplified by the high-fidelity PCR system (Toyobo, Osaka, Japan), and the primer sequences used were shown in Table 4. According to the instruction, JUNO mRNA was transcribed in vitro using T7 High Yield RNA Synthesis Kit (NEB, Beijing, China) for 4 h at 37 °C with 1–2 μg template and 7.5 mM ATP, CTP, UTP, and GTP, followed by DNase treatment for 15 min. Then mRNA was purified with Trizol Reagent (Invitrogen, Carlsbad, CA, USA).

Western blot was performed to validate the function of JUNO mRNA. JUNO mRNA was transfected into 293T cells with a lipo2000 transfection reagent. After 48 h, the total protein was extracted from the 293T cells with RIPA buffer containing protease cocktail inhibitors and quantified with a BCA kit (Beyotime, Shanghai, China). The total protein was separated with SDS-PAGE electrophoresis and then transferred to a PVDF membrane. The PVDF membrane was blotted with PBST-5%BAS and incubated with the primary antibody anti-Flag (MA1–91878, 1:2000, ThermoFisher, Waltham, MA, USA). Membranes were then incubated with peroxidase-conjugated mouse secondary antibody 1:5000 (Cwbio, Beijing, China) and visualized on ChemiDoc using an ECL kit (GE Healthcare, Little Chalfont, UK).

### 4.7. The Assessment of Sperm Binding Ability

The oocytes were treated with acidic Tyrode’s solution (pH 2.5) for 20 s to remove zona pellucida (ZP). Before fertilization, ZP-free oocytes were placed in 10 μg/mL Hoechst-33342 for 5 min. If the fusion of membranes of sperm and oocyte occurred, the fluorescent signal in the sperm head increased due to the freely diffusing of Hoechst-33342 in the oocyte plasma membrane [97].

### 4.8. The Assessment of Normal Fertilization

After fertilization, the presumed oocytes were treated with 0.5% (*w*/*v*) pronase to remove the ZP, stained with 10 μg/mL Hoechst-33342 for 5 min, and washed twice. Then, oocyte images were obtained by a fluorescence microscope (Nikon, Tokyo, Japan) to observe fertilization. Oocytes were classified into three groups according to the pronuclear formation: (1) normal fertilization: the presence of two pronuclei in the ooplasm, (2) unfertilized oocytes: the absence of pronuclei in the ooplasm, and (3) polyspermic fertilization: the presence of more than two pronuclei in the ooplasm.

### 4.9. Microinjection of JUNO mRNA

The COCs were randomly segregated into four groups and injected with 100 pg, 200 pg, and 300 pg JUNO mRNA or DEPC-treated water, respectively. The oocytes were mixed in a 200 μL drop of an IVM medium overlaid with mineral oil, and the microinjection needle was adjusted to 30° and loaded with JUNO mRNA or DEPC-treated water. The microinjection was performed under a Nikon inverted microscope (Nikon, Tokyo, Japan) equipped with a microinjector apparatus (Eppendorf, Hamburg, Germany). After injection, they were cultured in an IVM medium for maturation, and only MII oocytes were used for the experiment.

### 4.10. The Preparation and Addition of CLC/MβCD

According to the description by Purdy [98], MβCD was chosen as a carrier for CLC addition or removal. Briefly, 200 mg CLC was added to 1 mL of chloroform to form a solution, and then 0.45 mL of the cholesterol solution was mixed with the 2 mL methanol containing 1 g MβCD, to prepare the CLC/MβCD. After thoroughly mixing the solution, the solvents were removed with nitrogen gas. The resulting crystals were dried for 24 h and stored at 22 °C in a glass container. Obtained crystals were dried for 24 h and kept in glassware at 22 °C, until prepared as a solution, according to experimental requirements. Oocytes were incubated with 15 mM CLC at 38.5 °C for 45 min before vitrification, and then incubated with 4.25 mM MβCD at 38.5 °C and 5% CO_2_ for 45 min after vitrification, to remove the extra CLC from the oocyte plasma membrane.

### 4.11. Analysis of Phosphatidylserine (PS) Externalization Events

The Annexin V-FITC apoptosis detection Kit (Beyotime, Shanghai, China) was used to detect the PS externalization of oocytes. In brief, oocytes were washed with 1 × Annexin V binding buffer three times and then incubated in 5 μL Annexin V and 5 μL PI at room temperature, for 15 min in the dark. After staining, oocytes were washed twice with a binding buffer. The fluorescent images were collected with an inverted microscope (TE2000-U; Nikon, Tokyo, Japan) connected to a DS-Ri1 camera (Nikon, Tokyo, Japan).

According to the method of Anguita [99], bovine oocytes were divided into three groups after Annexin V staining, representing (1) intact oocytes with no Annexin V staining, (2) early apoptotic oocytes with Annexin V-positive signal in the membrane, and (3) necrotic oocytes with PI-positive red nuclei.

### 4.12. Quantitative Real-Time PCR of Candidate Genes

The total RNA was obtained from samples by TriZol reagent (Invitrogen, Carlsbad, CA, USA), dissolved in sterile water, and stored at −80 °C until use. The quantitative real-time PCR procedure was performed on an ABI 7500 SDS instrument (Applied Biosystems, Waltham, MA, USA) with the comparative Ct (2^−ΔΔ^Ct) method [100]. The *β-ACTIN* was chosen as the reference gene. Table 5 listed the primers used in the present study.

### 4.13. Total Nuclear Counts of Blastocysts

The blastocysts were incubated in 5 mg/mL pronase to remove the ZP and then incubated with 10 μg/mL Hoechst-33342 for 5 min. Subsequently, they were mounted on a slide and measured by a fluorescence microscope camera equipped (Olympus, Tokyo, Japan) with a CoolSNAP HQ CCD.

### 4.14. Experiment Design

LC-MS/MS was utilized to determine JUNO protein sequences and post-translational modifications. JUNO protein abundance, sperm binding capacity, fertilization capacity, and developmental capacity of vitrified bovine oocytes after JUNO mRNA microinjection were measured. Subsequently, the membrane integrity, sperm binding ability, fertilization ability, and developmental ability of vitrified oocytes treated with JUNO mRNA microinjection and CLC/MβCD were determined. Additionally, the gene expression levels of *IFN-tau*, *BCL2*, *OCT4*, *CDX2*, *ATP6*, *BAX*, and *SOD1* were analyzed to assess the quality of oocytes or embryos.

### 4.15. Statistical Analysis

All experiments were repeated at least three times and the results were expressed as mean ± standard error. The arcsine transformation of the percentages was performed before analysis. The SAS software 8.0 (SAS Institute, Carrey, NC, USA) was used to perform the one-way analysis of variance (ANOVA). Values with *p* < 0.05 were considered statistically significant.

## 5. Conclusions

In conclusion, our results show that vitrification of bovine oocytes did not alter the protein sequence of JUNO, but induced post-translational modifications and changed protein abundance. The combination treatment of JUNO mRNA + CLC/MβCD could protect the fertilization capacity and development ability of vitrified bovine oocytes by maintaining the normal abundance and function of JUNO protein.

## Figures and Tables

**Figure 1 ijms-24-00590-f001:**
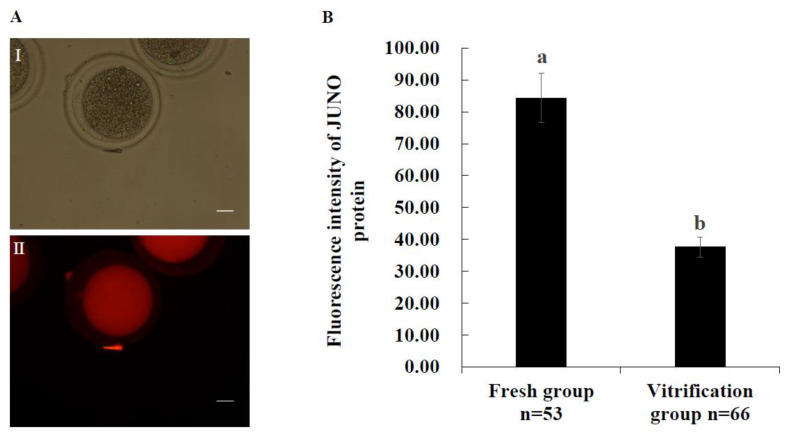
Effect of vitrification on JUNO protein abundance in bovine oocytes. (**A**): Representative images of JUNO staining. (**I**) The white light of the oocyte. (**II**) The image of JUNO staining. Scale bar = 20 μm. (**B**): Effect of vitrification on JUNO protein abundance in oocytes. Values with no common superscript lowercase letters mean significant differences between groups (*p*  <  0.05).

**Figure 2 ijms-24-00590-f002:**
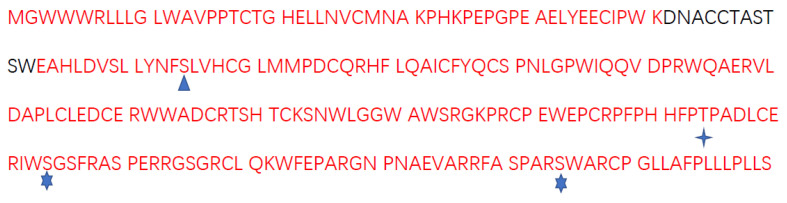
Effects of vitrification on sequences and post-translational modifications of JUNO protein. Sequences marked in red were the sequences detected in this experiment, and sequences marked in black were the undetected sequences. 
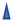
: The phosphorylation site was detected in only the vitrification group. 
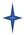
: The glycosylation site was detected in both the fresh group and the vitrification group. 
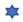
: The glycosylation sites were detected in only the vitrification group.

**Figure 3 ijms-24-00590-f003:**
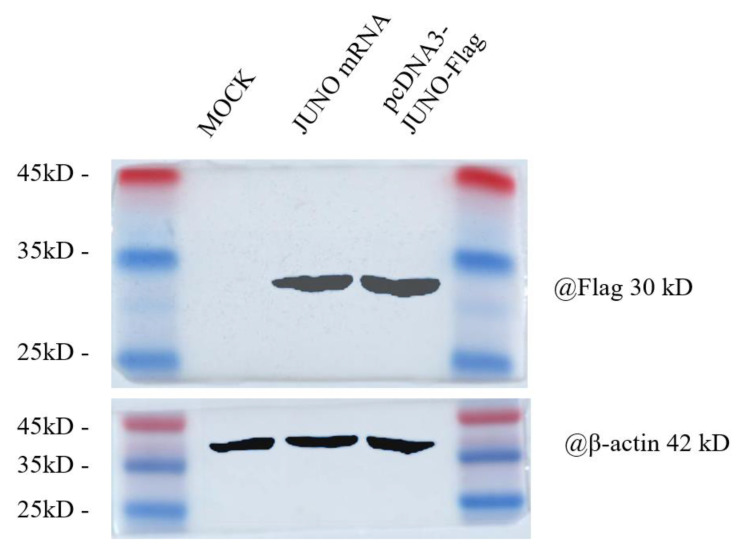
Western blot of JUNO protein in 293T cells.

**Figure 4 ijms-24-00590-f004:**
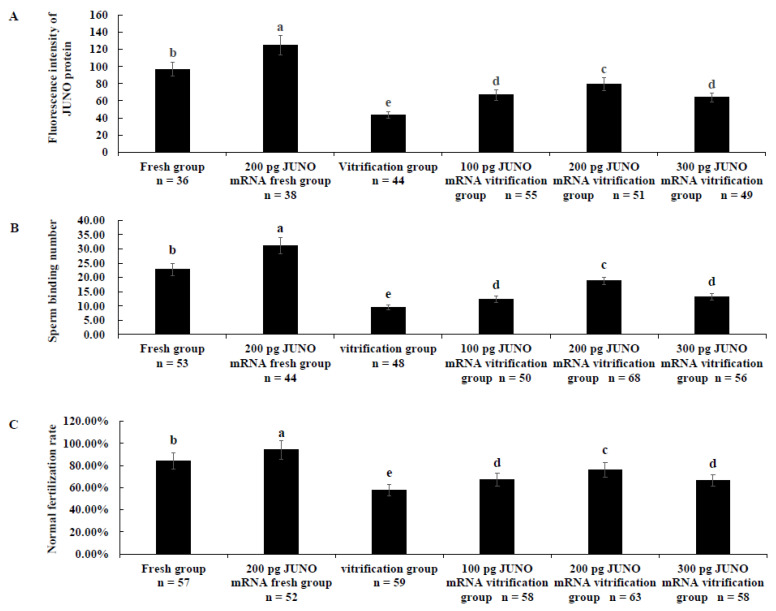
Effect of JUNO mRNA microinjection on the abundance of JUNO and fertilization ability of oocytes. (**A**): Effect of JUNO mRNA microinjection on the protein abundance of JUNO of vitrified oocytes. (**B**): Effect of JUNO mRNA microinjection on the sperm binding capacity of vitrified oocytes. (**C**): Effect of JUNO mRNA microinjection on the normal fertilization rate of vitrified oocytes. Values with no common superscript lowercase letters mean significant differences between groups (*p*  <  0.05).

**Figure 5 ijms-24-00590-f005:**
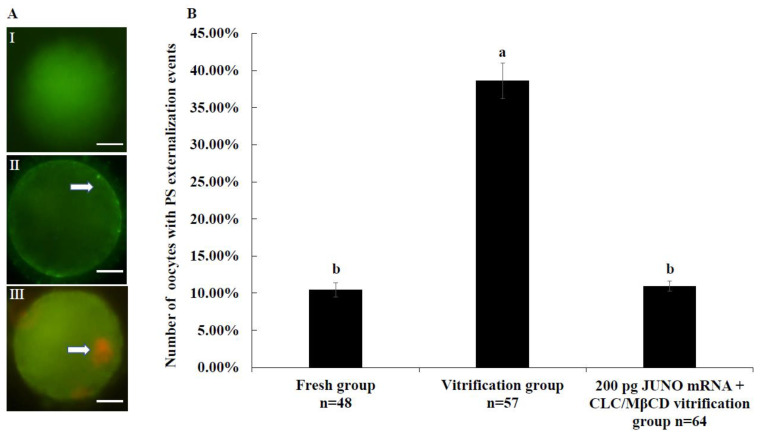
Effect of JUNO mRNA + CLC/MβCD treatment on the membrane integrity of vitrified oocytes. (**A**): Representative images of Annexin V staining, scale bar = 20 μm. (**I**) The image of Annexin V-negative oocyte. (**II**) The image of Annexin V-positive oocyte, the white arrow indicates the apoptotic signal. (**III**) The image of necrotic oocyte, the white arrow indicates necrotic nuclei. (**B**): Effect of JUNO mRNA + CLC/MβCD treatment on the PS externalization of vitrified oocytes. Values with no common superscript lowercase letters mean significant differences between groups (*p*  <  0.05).

**Figure 6 ijms-24-00590-f006:**
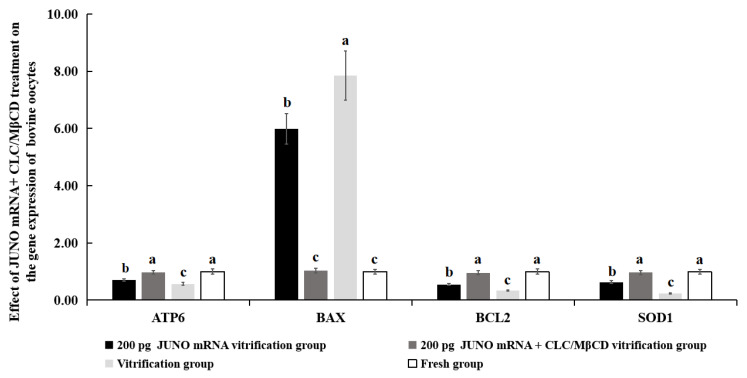
Effect of JUNO mRNA + CLC/MβCD treatment on the gene expression of bovine oocytes. Values with no common superscript lowercase letters mean significant differences between groups (*p*  <  0.05).

**Figure 7 ijms-24-00590-f007:**
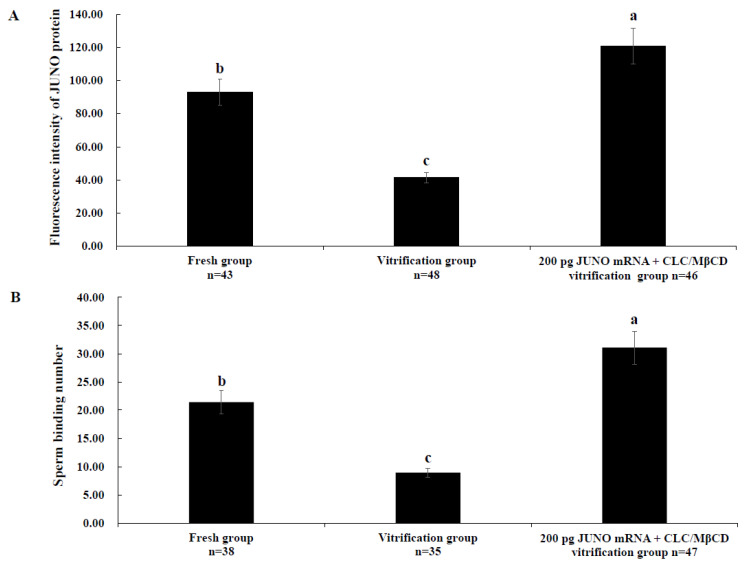
Effect of JUNO mRNA + CLC/MβCD treatment on the JUNO abundance and sperm binding number of oocytes. (**A**): Effect of JUNO mRNA + CLC/MβCD treatment on the JUNO protein abundance of vitrified oocytes. (**B**): Effect of JUNO mRNA + CLC/MβCD treatment on the sperm binding number of vitrified oocytes. Values with no common superscript lowercase letters mean significant differences between groups (*p*  <  0.05).

**Figure 8 ijms-24-00590-f008:**
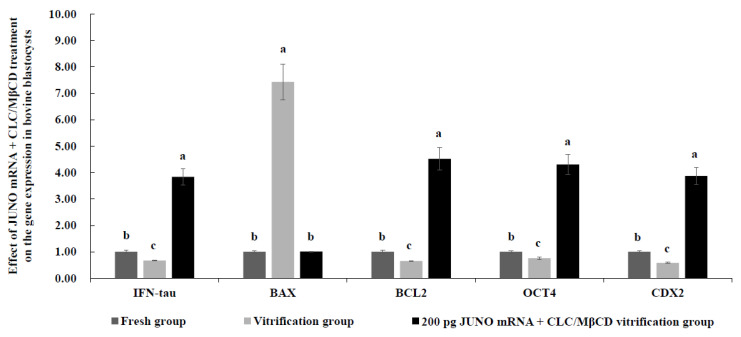
Effect of the JUNO mRNA + CLC/MβCD treatment on the gene expression in bovine embryos. Values with no common superscript lowercase letters mean significant differences between groups (*p*  <  0.05).

**Table 1 ijms-24-00590-t001:** Effects of vitrification on disulfide bond modification site of JUNO protein.

Groups	Site	Peptide	Protein Type
Fresh group	18–179	AVPPTCTGHELL (6)-PTPADLCERIW (7)	Intra
	18–165	AVPPTCTGHELL (6)-EPCRPF (3)	Intra
	27–125	APLCLEDCE (4)-LLNVCMNAK (5)	Intra
	79–47	SLVHCGL (5)-EECIPW (3)	Intra
	79–79	SLVHCGL (5)-SLVHCGL (5)	Inter
	79–99	SLVHCGLMM (5)-YQCSPNL (3)	Intra
	95–165	LQAICFY (5)-EPCRPF (3)	Intra
	99–165	EPCRPFPHHF (3)-QCSPNL (2)	Intra
Vitrification group	79–79	NFSLVHCGLM (7)-NFSLVHCGLM (7)	Inter
	79–179	PTPADLCERIW (7)-SLVHCGLMM (5)	Intra
	79–229	ARCPGLLAFPL (3)-NFSLVHCGL (7)	Intra
	95–95	LQAICFY (5)-LQAICFY (5)	Inter
	125–142	VLDAPLCLEDCE (7)-TSHTCKC (5)	Intra

**Table 2 ijms-24-00590-t002:** Effect of JUNO mRNA microinjection on the development ability of vitrified bovine oocytes after IVF.

Groups	No. of Oocytes	No. of Cleavage Embryos (%)	No. of Blastocysts (%)
Fresh group	126	108 (85.71 ± 7.83%) ^b^	49 (45.37 ± 4.09%) ^b^
200 pg JUNO mRNA fresh group	119	115 (96.64 ± 8.17%) ^a^	65 (56.52 ± 4.87%) ^a^
Vitrification group	132	72 (54.55 ± 4.39%) ^d^	8 (11.11 ± 1.02%) ^d^
200 pg JUNO mRNA vitrification group	145	102 (70.34 ± 5.92%) ^c^	27 (26.47 ± 2.45%) ^c^

Values with no common superscript lowercase letters mean significant differences between groups (*p* < 0.05).

**Table 3 ijms-24-00590-t003:** Effect of JUNO mRNA + CLC/MβCD on the development ability of vitrified bovine oocytes after IVF.

Groups	No. of Oocytes	No. of Cleavage Embryos (%)	No. of Blastocysts (%)	Cell Number Per Blastocyst
Fresh group	224	193 (86.16 ± 7.34%) ^a^	83 (43.01 ± 3.08%) ^a^	102.32 ± 9.02 (n = 30) ^a^
Vitrification group	675	385 (57.04 ± 4.92%) ^b^	50 (12.99 ± 1.12%) ^b^	80.87 ± 7.38 (n = 30) ^b^
200 pg JUNO mRNA + CLC/MβCD vitrification group	284	228 (80.28 ± 4.92%) ^a^	91 (39.91 ± 3.27%) ^a^	99.03 ± 8.38 (n = 30) ^a^

Values with no common superscript lowercase letters mean significant differences between groups (*p * <  0.05).

**Table 4 ijms-24-00590-t004:** Primer sequences for high fidelity PCR.

Genes	Primer Sequences (5′-3′)
*JUNO*	T7 pro F: TAATACGACTCACTATAGGG BGH-rev: TAGAAGGCACAGTCGAGG

**Table 5 ijms-24-00590-t005:** Primer used in the present experiment.

Genes	Primer Sequences (5′-3′)	References
*JUNO*	F: GGTTCACTGCGGACTAATGATG R: GGGAGCACTGGTAGAAGCAGAT	Hao [34]
*ATP6*	F: GAACACCCACTCCACTAATCCCAAT R: GTGCAAGTGTAGCTCCTCCGATT	Zhao [81]
*BAX*	F: TTTCTGACGGCAACTTCAACTG R: GGTGCACAGGGCCTTGAG	Zhao [101]
*BCL2*	F: TCAATTGTCGTGGCATCAAAA R: CCCCCGACACCTGTTAGCTT	Zhao [101]
*SOD1*	F: CATCCACTTCGAGGCAAAGG R: GGTCTCCAACATGCCTCTCT	Zhao [69]
*IFN-tau*	F: GCTCCAGAAGGATCAGGCTATC R: TGTTCCAAGCAGCAGACGAGT	Zhao [81]
*OCT4*	F: CCACCCTGCAGCAAATTAGC R: CCACACTCGGACCACGTCTT	Su [102]
*CDX2*	F: GCAAAGGAAAGGAAAATCAACAA R: GGGCTCTGGGACGCTTCT	Su [102]
*β-ACTIN*	F: TCCTGGGCATGGAATCCTG R: GGCGCGATGATCTTGATCTTC	Zhao [101]

## Data Availability

The datasets supporting the conclusions of this article are included within the article and the Appendix A.

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
