# Peer review of "Improvement of Fertilization Capacity and Developmental Ability of Vitrified Bovine Oocytes by JUNO mRNA Microinjection and Cholesterol-Loaded Methyl-β-Cyclodextrin Treatment"

_ijms, 2022, doi:10.3390/ijms24010590_

Round 1
Reviewer 1 Report
I accept the manuscript in its present form. The Authors comprehensively present the aims of the research. The results confirm the hypothesis about the improvement of fertilization capacity by Juno mRNA microinjection. I recommend this article for publication in IJMS.
Author Response
Question 1: English language and style are fine/minor spell check required.
ANSWER: Thanks for your suggestions. We have invited native English speaker to polish our paper.
Reviewer 2 Report
The here presented work combine different technical approaches to study the alteration and restoration of JUNE protein in vitrified bovine oocytes. Vitrification did not alter the amino acid sequence of JUNO protein, but led to changes in post-translational modifications. An interesting contribution is that the combination treatment of JUNO mRNA + CLC/MβCD 540 could protect the fertilization capacity and development ability of vitrified bovine oocytes by maintaining the normal level and function of JUNO, as measured to the blastocyst stage within in vitro culture. The work is interesting, technically correct and understandable.
The manuscript needs re-writing.
Discussion needs shortening
ABSTRACT should briefly explain why JUNO is important for fertilization of vitrified oocytes (i.e., the hypothesis).
L21-23: Reformulate the text as follows: Our results showed that vitrification of bovine oocytes did not alter the protein sequence of JUNO, but induced post-translational modifications and changed protein abundance
L23-25: Poorly written sentence
L30: “deprive” is misused.
L33-34: “in situ conservation” and “gene banking” are terms difficult to match.
L53: There is an author name in capitals, which is incorrect.
L61: What the authors wish to explain with “level” (concentration, abundance?) and where (i.e., in the oocyte itself, extracellular?).
L114: “a previous method” / “examined” better than “detected”.
L120-121: The fields, surface and/or parameters used within the JUNO quantification process must be defined.
L135: “performing” the disulfide bonds is unclear. Please, define.
L136-137: Briefly define the Byonic score.
L166-167: Incorrect sentence
L240: Imagines / images
L302-307: Avoid to repeat as text the numbers that can be read in tables.
L389-391: Sentence out of the focus (it is obvious, as probably many more proteins)
L415-418: The authors did not study effects of vitrification in sperm; please, remove wherever.
Figure 2
Explain what the different symbols indicate.
L271-272: this sentence needs reformulation and clarity
Figure 6
Needs color in the bars and in the squares of legends (i.e., the reading order -and the representativity of squares- is confusing in legends)
Author Response
Reply to Referee 2
Comments and Suggestions for Authors
The here presented work combine different technical approaches to study the alteration and restoration of JUNO protein in vitrified bovine oocytes. Vitrification did not alter the amino acid sequence of JUNO protein, but led to changes in post-translational modifications. An interesting contribution is that the combination treatment of JUNO mRNA + CLC/MβCD could protect the fertilization capacity and development ability of vitrified bovine oocytes by maintaining the normal level and function of JUNO, as measured to the blastocyst stage within in vitro culture. The work is interesting, technically correct and understandable.
Question 1: The manuscript needs re-writing.
ANSWER: Thanks for your suggestions. We have invited native English speaker to polish our paper.
Question 2: Discussion needs shortening
ANSWER: Thanks for your suggestions and we have done our best to trim the sentences in the discussion section.
Question 3: ABSTRACT should briefly explain why JUNO is important for fertilization of vitrified oocytes (i.e., the hypothesis).
ANSWER: Thanks for your suggestions. We have explained briefly why JUNO is important for fertilization of vitrified oocytes and details are shown in the L13-16.
Juno protein, as a receptor for Izumo1, is involved in sperm-oocyte fusion and is an indispensable protein for mammalian fertilization, and its abundance is susceptible to vitrification, aging and drugs. However, it is still unclear how vitrification reduces the fertilization capacity of bovine oocytes by affecting JUNO protein.
Question 4: L21-23: Reformulate the text as follows: Our results showed that vitrification of bovine oocytes did not alter the protein sequence of JUNO, but induced post-translational modifications and changed protein abundance.
ANSWER: Thanks for your suggestions and we have replaced “our results showed that vitrification did not alter the protein sequence of JUNO protein but significantly changed the protein level and post-translational modifications in bovine oocytes” with “our results showed that vitrification of bovine oocytes did not alter the protein sequence of JUNO, but induced post-translational modifications and changed protein abundance”. The details are shown in the L26-28.
Question 5: L23-25: Poorly written sentence
ANSWER: Thanks for your suggestions. We have done the revised and the details are shown in the L28-31.
Moreover, the fertilization and development ability of vitrified bovine oocytes were improved by the combination treatment of JUNO mRNA microinjection and CLC/MβCD.
Question 6: L30: “deprive” is misused.
ANSWER: Thanks for your suggestions. We have replaced “deprive” with “break” and details are shown in the L36.
Oocyte cryopreservation provides sufficient materials for animal biotechnologies to break the limitation of time and space.
Question 7: L33-34: “in situ conservation” and “gene banking” are terms difficult to match.
ANSWER: Thanks for your suggestions and we have replaced “in situ conservation” with “protecting the living organism” and added “to storage genetic resources”. The details are shown in the L39-40.
Moreover, oocyte cryopreservation provides the opportunity to keep the safety of oocytes from epidemic diseases [6] and it is more convenient and economical than protecting the living organism [7], which can be used for gene banking of female animals to storage genetic resources.
Question 8: L53: There is an author name in capitals, which is incorrect.
ANSWER: Thanks for your suggestions. We have changed “INOUE” to “Inoue”and the details are shown in the L60.
However, Inoue [27] reported that the binding of Izumo1 to the surface of oocytes may not need CD9, and oocytes lacking CD9 can be fertilized, which suggests that CD9 is not the only receptor for Izumo1.
Question 9: L61: What the authors wish to explain with “level” (concentration, abundance?) and where (i.e., in the oocyte itself, extracellular?).
ANSWER: Thanks for your suggestions. We wish to use “level” to explain “abundance” of JUNO in oocytes. We have changed the “level” to “abundance” in the whole article. The details are shown in the L68, L72, L78 and so on.
Question 10: L114: “a previous method” / “examined” better than “detected”.
ANSWER: Thanks for your suggestions and we have done the revised as following:
We have changed the “the previous method” to “a previous method”. We have changed the “detected” to “examined”. The details are shown in the L122-123.
According to a previous method [22], the abundance of JUNO protein in oocytes was examined.
Question 11: L120-121: The fields, surface and/or parameters used within the JUNO quantification process must be defined.
ANSWER: Thanks for your suggestions and we have described the method of JUNO quantification as shown in the L129.
Finally, the epifluorescence inverted microscope (TE2000-U; Nikon, Tokyo, Japan) connected to a DSRi1 camera (Nikon) was used to take images of oocytes along the equatorial plane of the oocyte at 10X magnification.
Question 12: L135: “performing” the disulfide bonds is unclear. Please, define.
ANSWER: Thanks for your suggestions and we have changed “performing” to “analyzed” as shown in the L145.
The disulfide bonds of JUNO protein were analyzed using pLink.
Question 13: L136-137: Briefly define the Byonic score.
ANSWER: Thanks for your suggestions and we have described “Byonic score” as shown in the L145-147.
The phosphorylation and glycosylation were analyzed by Byonic. JUNO protein's phosphorylation and glycosylation sites were selected only with a Byonic score ≥100 because a minimum Byonic score of 100 is considered as highly confident assignment [38].
Question 14: L166-167: Incorrect sentence
ANSWER: Thanks for your suggestions and we have done the revised as shown in the L180-181.
Then, oocyte images were obtained by a fluorescence microscope (Nikon) to observe fertilization.
Question 15: L240: Imagines / images
ANSWER: Thanks for your suggestions and we have changed “imagines” to “images” as shown in the L251.
A: Representative images of JUNO staining.
Question 16: L302-307: Avoid to repeat as text the numbers that can be read in tables.
ANSWER: Thanks for your suggestions and we have deleted the numbers that can be read in tables. The details are shown in the L315-318 and L364-368.
Question 17: L389-391: Sentence out of the focus (it is obvious, as probably many more proteins)
ANSWER: Thanks for your suggestions and we have removed references to other proteins as shown in the L460-461.
Question 18: L415-418: The authors did not study effects of vitrification in sperm; please, remove wherever.
ANSWER: Thanks for your suggestions and we have removed the sperm and modified the sentence to make it easier to understand. The details are shown in the L433-436.
Besides, phosphorylation of tyrosine residues of proteins is important for activities related to fertilization [73], such as ZP binding and the binding and fusion of sperm-oocyte [73, 74].
Question 19: Figure 2 :Explain what the different symbols indicate.
ANSWER: Thanks for your suggestions and we have explained what the different symbols indicate as shown in the L274-276.
Figure 2. Effects of vitrification on sequences and post-translational modifications of JUNO protein. Sequences marked in red were the sequences detected in this experiment, and sequences marked in black were the undetected sequences.
: The phosphorylation site was detected in only the vitrification group.
:The glycosylation site was detected in both the fresh group and vitrification group.
:The glycosylation sites were detected in only the vitrification group.
Question 20: L271-272: this sentence needs reformulation and clarity
ANSWER: Thanks for your suggestions and we have redescribed as shown in the L279-280.
The prepared JUNO mRNA was transfected into 293T cells and the abundance of JUNO protein in 293T cells was verified by western blot analysis.
Question 21: Figure 6:Needs color in the bars and in the squares of legends (i.e., the reading order -and the representativity of squares- is confusing in legends)
ANSWER: Thanks for your suggestions and we have modified as shown in the L345-346.
